# Degradation of Crystal Violet by Catalytic Wet Peroxide Oxidation (CWPO) with Mixed Mn/Cu Oxides

**Ana María Campos [1], Paula Fernanda Riaño [1], Diana Lorena Lugo [1], Jenny Alejandra Barriga [1], Crispín Astolfo Celis [1], Sonia Moreno [2] and Alejandro Pérez [1,\*]**

[1] Línea de Investigación en Tecnología Ambiental y de Materiales (ITAM), Departamento de Química, Facultad de Ciencias, Pontificia Universidad Javeriana, Carrera 7 No. 43-82, Bogotá D.C., P.C. 110231, Colombia; amcamposr@yahoo.es (A.M.C.); paula.riano2@estudiantes.uamerica.edu.co (P.F.R.); d.moralesl@javeriana.edu.co (D.L.L.); barriga_j@javeriana.edu.co (J.A.B.); crispin.celis@javeriana.edu.co (C.A.C.)

[2] Estado Sólido y Catálisis Ambiental (ESCA), Departamento de Química, Facultad de Ciencias, Universidad Nacional de Colombia, Carrera 30 45-03, Bogotá, P.C. 110231, Colombia; smorenog@unal.edu.co

\* Correspondence: alejandroperez@javeriana.edu.co; Tel.: +571-3208-320 (ext. 4124)

**Abstract:** The environment protection has been the starting point for the development of new technologies, which allow the control of highly toxic substances present in the effluents of various industries, whose removal is not feasible by conventional methods. In this research, mixed oxide catalysts Mn and Cu in different molar ratios were prepared from the autocombustion method and characterized by XRD, XRF, TPR-$H_2$, and $N_2$ adsorption–desorption isotherms. The solids were evaluated in the catalytic wet peroxide oxidation of crystal violet (CV) with mild conditions of reaction: 25 °C, normal pressure, airflow of 2 mL/min, and $H_2O_2$ 0.1 M (2 mL/h). The experimental results indicated degradations of 100% of CV, conversion of the total organic carbon (TOC) of 74%, and elimination of chemical oxygen demand (COD) of 71% in 90 min of reaction. Additionally, the selectivity was monitored by CG-MS, finding that there was almost complete mineralization in a short reaction time, generating intermediate products such as carboxylic acids, alcohols, and amines that do not cause a serious risk to the environment. The Mn–Cu catalyst with molar ratios of 1:2 was the most promising catalyst, displaying a cooperative effect between the two metals, and demonstrating the importance of the redox properties for the elimination of CV dye in wastewater.

**Keywords:** catalytic wet peroxide oxidation; degradation; autocombustion; crystal violet

## 1. Introduction

The aim of wastewater treatment in industry is to reduce the load of pollutants from water and meet the discharge regulations. Wastewaters containing high concentrations of persistent, toxic, and nonbiodegradable organic pollutants (e.g., aromatics, pesticides, etc.) are hard to treat with conventional physical–chemical or biological methods.

Water treatment recalcitrant and toxic residuals have been studied because of their danger with various methods used in order to obtain efficiency and profitability at the industrial level [1]. Within the wastewater emanating from textile, leather, plastic, and pharmaceutical companies there are dissolved substances that generate an increase in the chemical oxygen demand (COD) due to organic chemical matter not being biodegradable. The synthetic dye crystal violet, used often in staining processes, histological staining, as a dermatological agent, and in forensic medicine, has been cataloged as a highly toxic cationic dye, even at very low concentrations, susceptible to oxidation reactions and

hydrolysis, where toxic metabolites are produced in waters, which generate adverse effects in both animals and human health [2,3].

According to the multiple adverse effects of crystal violet (CV) as a carcinogenic and mutagenic agent, different methods, including adsorption techniques, bioremediation, photolysis, and photocatalysis, have been studied for removing water residuals, taking into account that many of these show limitations, such as a high cost for manipulation of conditions such as pressure and temperature, processes subsequent separation due to generated byproducts, and low efficiency [4]. On the other side, the advanced processes of oxidation (APOs) arises as one of the most promising alternatives, with high yields obtained; in which oxidation with hydrogen peroxide (CWPO: catalytic wet peroxide oxidation) is highlighted because it uses heterogeneous catalysts with low cost and easy handling [5]. The axis fundamental to the process is the reaction of Fenton, where hydrogen peroxide and $Fe^{+2}$ salts are mainly used as reagents for the formation of the radical hydroxyl (HO·), being the active intermediate that facilitates the elimination of contaminants in the effluents [6,7]. The behavior of the CWPO reaction using salts of iron is very susceptible to the values of pH [8,9]. According to some authors, the maximum speed of elimination of this reaction is obtained in a range of pH values between 2.5 and 3.5, which coincides with the interval in which the decomposition of hydrogen peroxide is minimal [10]. Because of this, studies have been carried out that allow an extension of the pH range by implementing others metals such as Cu [11], presenting an advantage in wastewater treatment where pH ranges from around 6 to 9, which is characteristic for an alkaline medium.

CWPO has been employed as both a homogeneous and a heterogeneous catalyst, seeking to improve conditions of pressure, temperature, and residence time. However, an oxidation process that can employ a solid catalyst greatly facilitates the decontamination process [12]. The mixed oxides show large advantages as catalysts, due to the textural, morphological, structural, and catalytic properties conferred within the crystalline structure that characterizes them with high superficial areas, good stability, and activity [13]. Their synthesis is by numerous methods, such as hydrothermal, coprecipitation, sol-gel, autocombustion, and microemulsion impregnation, in which more efficient molecular structures are sought for the oxidation mechanisms involved [14]. Within these methods, the autocombustion method allows synthesis in short times, does not need intermediaries, and is simple and versatile to implement. The solids obtained have a wide range of particle sizes, crystallinity, and high purity [15,16].

Regarding the metal used for CWPO, the transition metal oxides Cu, Ni, Co, and Mn have advantages among other metals due to their easy availability and low cost. Concerning their oxides, Cu and Mn oxides are the most active potential candidates for catalytic oxidation processes. The catalytic properties of Mn are attributed to its ability to form oxides with different oxidation states, $Mn^{2+}/Mn^{3+}$ or $Mn^{3+}/Mn^{4+}$, and its oxygen storage capacity in the crystalline lattice [17].

Considering the latest aspects, in this work, mixed oxides of Mn and Cu were synthesized by the autocombustion method for the catalytic wet peroxide oxidation of violet crystal, with a more extended range of the pH than is normally employed in Fenton-type reactions. In addition, the evaluation of a possible cooperative effect within these transition metals was studied.

## 2. Results

### 2.1. Mixed Oxide Catalyst

In order to evaluate the degradation of CV, five solids were synthesized, defined by the following nomenclature: Mn, Cu, Mn–Cu (1:1), Mn–Cu (1:2), and Mn–Cu (2:1), with the stoichiometric ratio (Table 1).

**Table 1.** Stoichiometric ratio and nomenclature of oxide-mixed catalyst.

| Catalyst | Mn/Cu | Al | Mg | Glycine |
|----------|-------|----|----|---------|
| Mn–Cu (1:1) | 1 | 20 | 30 | 100 |
| Mn–Cu (1:2) | 0.5 | 20 | 30 | 100 |
| Mn–Cu (2:1) | 2 | 20 | 30 | 100 |
| Mn | 0 | 20 | 30 | 100 |
| Cu | 5 | 20 | 30 | 100 |

*2.2. Characterization of Catalyst*

Table 2 summarizes the results of the chemical and structural properties of mixed manganese–copper oxides. The X-ray diffraction (XRD) profiles (Figure 1) provided evidence for the presence of the husmanite phase ($Mn_3O_4$) and copper oxide (CuO) for the oxides that present as active phase Mn and Cu, respectively, and periclase (MgO) in all the oxides. The size of particle reduction of solid Mn after the incorporation of Cu (MnCu) revealed a possible effect of Cu distribution on manganese oxides, being more important the change in the size of particle in the MnCu (1:2) oxide.

**Table 2.** Particle size of the mixed solids, according to the crystalline phase determined from the XRD signals. Ratios determined by XRF.

| Catalyst | Crystaline Phase | FWHM | | 2θ | | Crystal Size (nm) * | | Ratios | | |
|----------|------------------|------|----|----|----|---------------------|----|--------|-------|-------|
| | | Mn | Cu | Mn | Cu | | | Cu/Al | Mn/Al | Mg/Al |
| Mn | $Mn_3O_4$ | 0.2525 | - | 44.4 | - | 6.59 | - | - | 0.4 | 1.8 |
| Cu | $Cu_2O$ | - | 0.8762 | - | 42.5 | - | 1.89 | 0.4 | - | 1.8 |
| Mn–Cu (1:1) | $Mn_3O_4$; $Cu_2O$ | 0.2800 | 0.3244 | 44.4 | 42.5 | 5.51 | 5.09 | 0.2 | 0.2 | 1.7 |
| Mn–Cu (2:1) | $Mn_3O_4$; $Cu_2O$ | 0.9153 | 0.8840 | 44.4 | 42.5 | 1.82 | 1.87 | 0.1 | 0.3 | 1.8 |
| Mn–Cu (1:2) | $Mn_3O_4$; $Cu_2O$ | 1.2066 | 1.2212 | 44.4 | 42.5 | 1.38 | 1.35 | 0.3 | 0.1 | 1.8 |

* Crystal size was determined with the angles 2θ of 44.35 and 42.5 for the manganese and copper metals, respectively. FWHM: full width at half maximum.

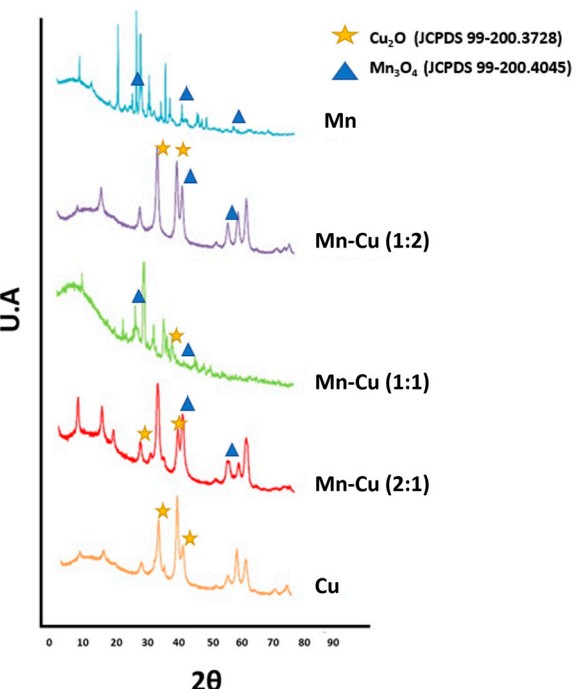

**Figure 1.** XRD patterns of catalyst.

Regarding the chemical composition by X-ray fluorescence XRF (Table 2) indicated through molar relationships between the elements, the catalysts revealed a similar and consistent chemical

composition with the nominal ratio used in the synthesis, highlighting that the autocombustion method allows solids to be obtained without loss of the nominal value.

Concerning the surface analysis, the solids had isotherms of type IV (IUPAC classification) with a hysteresis H3 type (Figure 2), which corresponds to the solids with rough surfaces that are in aggregates of particles in the form of plates and that give rise to pores in the form of slits [18,19].

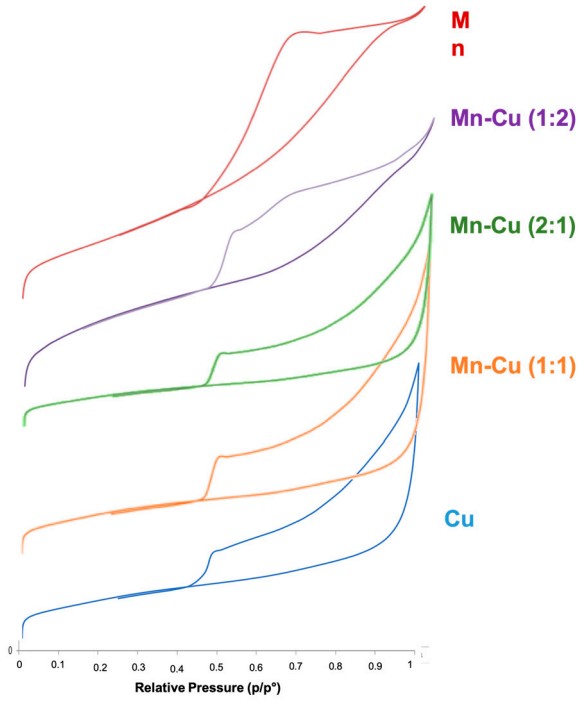

**Figure 2.** Nitrogen adsorption/desorption isotherms for catalyst.

BET areas are in Table 3. Table 3 also records the maximum temperature values of reduction in the analysis by TPR-$H_2$. Comparing the reduction temperatures of the solid Mn with the solids that have copper, the displacement at lower temperatures for mixed oxides MnCu indicated that reduction of manganese oxides is easier when Cu is added, with a further shift observed in the MnCu (1:2) oxide (Figure 3). This result can be related to the increased mobility of oxygen by the presence of a second metal of transition (Cu) and the particle size reduction that can facilitate the redox behavior in solids [15].

**Table 3.** Superficial area BET and maximum temperature values of reduction by TPR-$H_2$.

| Catalyst | BET Area [$m^2/g$] | T Max (°C) TPR-$H_2$ |
|---|---|---|
| Mn | 71 | 505 |
| Cu | 112 | 211 |
| Mn–Cu (1:2) | 92 | 367 |
| Mn–Cu (1:1) | 55 | 194 |
| Mn–Cu (2:1) | 62 | 412 |

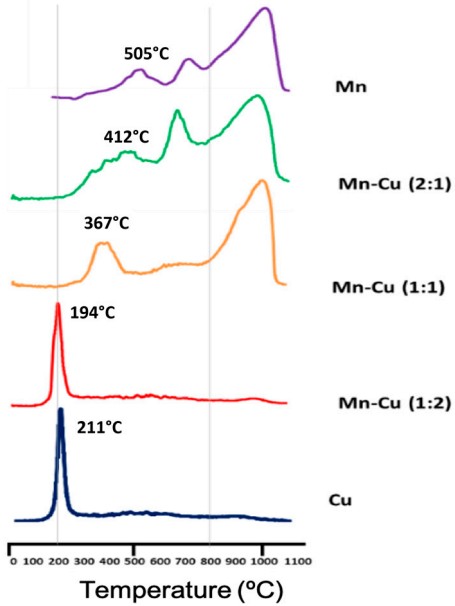

**Figure 3.** TPR-H$_2$ profiles for catalysts.

### 2.3. Catalytic Activity

Catalytic activity in CV oxidation is shown in Figure 4, where it can be appreciated that the activity was significantly influenced by the mixed oxide synthesis method, and a cooperative effect was observed between the two metals; the solid Cu–Mn (1:2) was the most active. These results are associated with the good redox properties of Mn and Cu, which are enhanced when they are together. Likewise, Figure 4 shows that the activity was directly related to the presence of a catalyst and not to the medium of reaction (H$_2$O$_2$ and air).

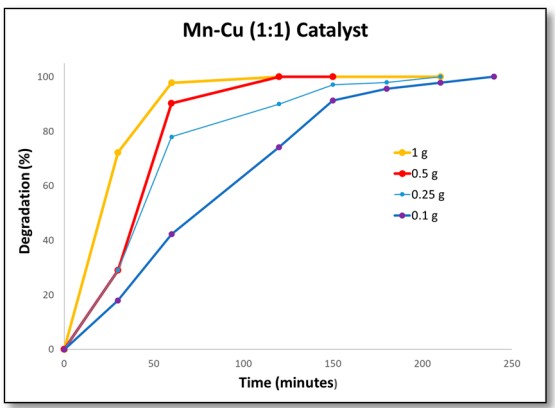

**Figure 4.** Effect of the catalyst loading in the catalytic wet peroxide oxidation (CWPO) of crystal violet (CV).

Additionally, with solid Mn–Cu (1:1), the effect of catalyst loading was evaluated (1, 0.5 and 0.25 g), finding as a result that only 0.25 g of catalyst yielded a degradation of 100% CV in 150 min (Figure 5), a selectivity in TOC of 70%, and an elimination in COD of 59% (Table 4). The results clearly indicated the great potential of these solids in respect to other catalysts reported in literature [20,21], under environmental conditions and low charges of active metal.

In order to clarify changes in the molecular and structural characteristics through CV oxidation as a function of time (Figure 5), GC-MS was used to further identify the intermediate products formed during the reaction (Figure 6). Equipment AGILENT 5975B VL MSD was used. At the beginning, the results showed evidence of the color degradation of the aromatic fragment in the molecule and the

appearance of different intermediaries due to breaks by ●OH radicals. Finally, the gradual breakdown of aromatic intermediate compounds led to the formation of carboxylic acids, before the conversion to carbon dioxide.

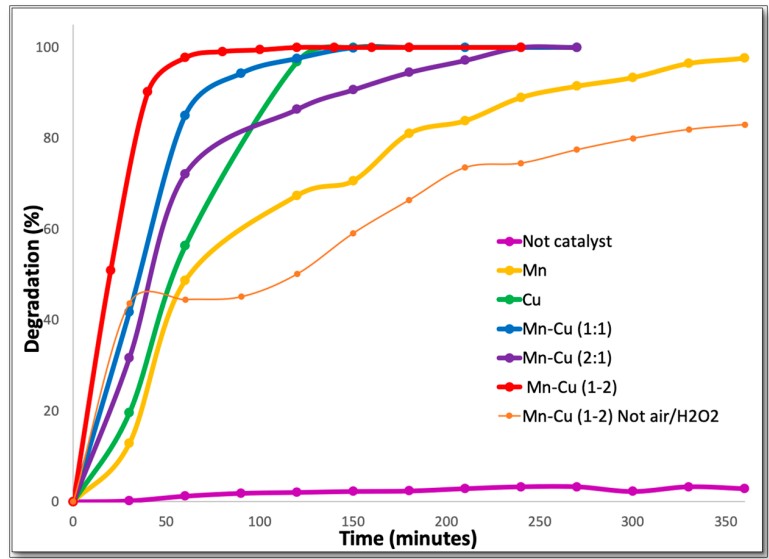

**Figure 5.** Catalytic activity for CWPO of CV by Mn–Cu catalyst.

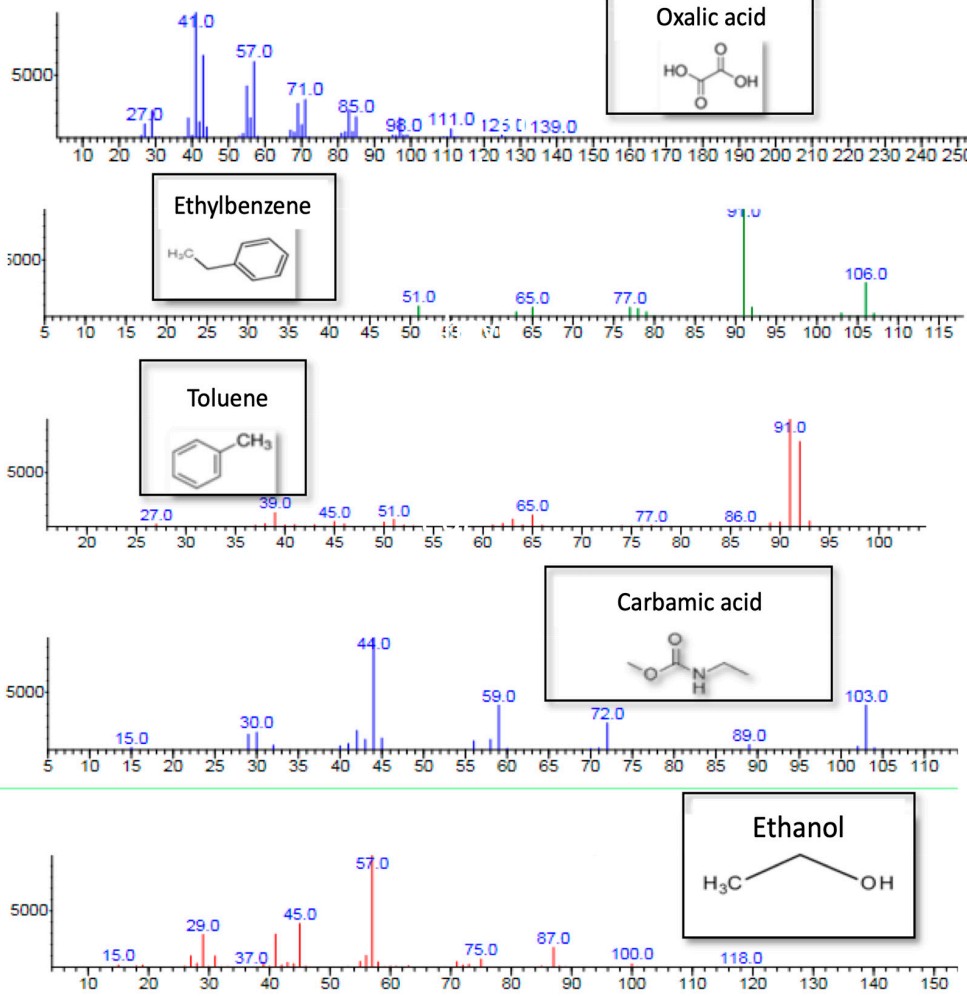

**Figure 6.** GC-MS chromatogram for CWPO in CV after 80 min of reaction.

**Table 4.** Catalytic activity in CV degradation of Mn–Cu catalyst.

| Catalyst | Mn | Cu | Mn–Cu | | |
| --- | --- | --- | --- | --- | --- |
| | | | 1:1 | 1:2 | 2:1 |
| Removal time 100% (min) | 360 | 150 | 150 | 60 | 240 |
| %TOC [1] (±2) | 49 | 68 | 70 | 74 | 52 |
| %COD [2] (±1) | 52 | 60 | 59 | 71 | 50 |

[1] % TOC: total organic carbon; [2] % COD: chemical oxygen demand.

Concerning lixiviation during the reaction process, at the end of the catalytic test, the stability of the active phases was evaluated by taking the solutions after the final reaction time, filtering them to separate the solid from the aqueous phase, and assessing the possible leaching of the catalyst metals during the reaction. The analysis was done by atomic absorption spectrophotometry technique using equipment Agilent 280.

The nonexistence of leaching of metals (Mn and Cu), measured by atomic adsorption, indicated that the active phase was stable under the reaction conditions and avoided the generation of additional pollutant [22].

## 3. Discussion

Regarding the XRD patterns (Figure 1), the non-formation of $Al_2O_3$ suggested that Al was part of the crystal lattice, and allowed better Cu dispersion as the active phase. Also, peaks observed in 2θ values of 44.6 and 66.3 corresponded to the spinel phase ($Al_2MgO_4$) and regarded the mixed oxides that were generated.

One of the principal signals for copper is in 2θ = 37.5 as the copper oxide (CuO) crystalline phase, followed by two minor peaks in 2θ = 36 and 42.5, which are attributed to the $Cu_2O$ cuprite (JCPDS 99-200-3728). These phases are essential in the Fenton oxidation process, since the Cu (II)/Cu (I) redox couples activate $H_2O_2$ molecules for the production of hydroxyl radical [23,24].

In contrast to Cu, the manganese pattern Mn (Figure 1) exposed different peaks associated with its multiple oxidation states and recorded signals corresponding to the periclase (MgO) phase. In addition, signals were related with spinel-type phases, which were preferential due to the high temperatures reached during the synthesis and subsequent calcination. This result showed the effectiveness of autocombustion as a novel method when compared with similar phases with traditional methods, such as co-precipitation [25].

Signals of $MnO_2$ at 2θ = 21, 28, and 36 are shown (Figure 1). MnO at 2θ = 26.8 and $Mn_3O_4$ at 2θ = 0.5 and 44.35 were related with these oxidative states as the most active phases. During the oxidation process, manganese oxide (III) and the husmanite phase $Mn_3O_4$ (JCPDS 99-200-4045) is highlighted as the most important phase in the decomposition of hydrogen peroxide during degradation processes [26–28].

The diffraction signals of the mixed catalyst of Mn–Cu (x:y) displayed the periclase phase and the formation of spinels, as occurred with the oxides separately. The crystalline phases that were related to the solids and standard signals evaluated previously, presented a favorable relationship with oxidizing agents such as oxygen and hydrogen peroxide for the production of radicals in the degradation of pollutants, evidenced by the most intense peaks belonging to $Cu_2O$, with values 2θ = 42.5 and 2θ = 44.35, respectively.

The increase of FWHM of XRD in mixed solids Mn–Cu (1:2) and Mn–Cu (2:1), with respect to Cu, Mn, and Mn–Cu (1: 1) solids, indicated that after the incorporation of Cu in the structure, a decrease in the crystallinity occurred. The latest result suggesting that the oxides are amorphous and highly dispersed structures. The decrease of the particle size of the solid of Mn, after the incorporation of the metal Cu (Mn–Cu), revealed a possible effect of distribution of cooper in the manganese oxides, being more significant than the change of the particle size in the oxide Mn–Cu (1:2).

The addition of Mn to the solids with copper led to lower surface areas being obtained, possibly due to processes of agglomeration of this metal on the surface. For the solid of Mn the particle size was greater in relation to the catalysts that present Cu, concluding that the adding of Cu as an active phase produced a decrease in particle size, which could result in a better distribution in the catalyst. The crystal size turned out to be better for the solid Mn–Cu (1:2), indicating high distribution on the surface of the active phase [29,30].

The average particle size (Table 2) was calculated from the Scherrer equation. For Mn solid, the particle size was greater in relation to the catalysts with Cu, so when adding Cu as an active phase there was a decrease in particle size, which could result in a better distribution in the catalyst. The smaller particle size of solid Mn–Cu (1:2) and Mn–Cu (2:1) indicated a good distribution of the active phase on the surface and therefore a possible better catalytic activity, which is evident in the reaction [27].

Regarding the results from the surface area BET, type IV isotherms characteristic of mesoporous solids were observed (Figure 2). In addition, according to the (IUPAC), the hysteresis type (H3) representative of a structure containing pores with different sizes and non-uniform shapes, as well as a large internal surface area, indicated the possible efficiency in the activity of the reaction [18,19]. The BET surface areas of the solids were summarized in Table 3. The Mn–Cu (1:2) sample had a surface area of 92 $m^2$/g, being smaller than the Cu solid (112 $m^2$/g), probably due to the partial obstruction of the pores and the addition of the active phase, where the introduction of Mn into the structure decreased the surface area and the pore volume of the material.

The resulting area was related to the evolution of gases such as $CO_2$ and $H_2O$, which were generated during the combustion process. Therefore, the solids with the greatest surface area were those with copper as the major phase. Thus, the mixed solid Mn–Cu (1:2) was the solid with the largest area compared to the other double-metal solids in the active phase.

According to the TPR-$H_2$ analysis (Figure 3), Cu catalyst showed a reduction peak between 200 °C and 300 °C, belonging to the reduction of cations from $Cu^{+2}$ to $Cu^0$, and another reduction peak above 450 °C. According to the literature, the peak of low temperature can be attributed to the reduction of cations from $Cu^{+2}$ to $Cu^0$, while the broad, but slightly pronounced peak is due to the hydrogenation of residual carbonates [31].

The Mn showed multiple peaks due to different oxidation states (Table 5) and finally, in binary samples Mn–Cu (x:y), comparing the reduction temperatures of the solid with only Mn or Cu, there was evidence of a shift to lower temperatures for the mixed oxides Mn–Cu, with a more pronounced shift in the Mn–Cu oxide (1:2) being observed, which indicated that the reduction of manganese oxides is easier when Cu is added (Table 3). This result can be attributed to the oxygen mobility increasing due to the presence of a second transition metal (Cu) and the decrease in particle sizes that can facilitate the redox behavior of the materials [32]. When Cu is added within the mixed oxides, lower reducibility temperatures are evident, which would favor the reaction to moderate conditions.

**Table 5.** Temperature range for reduction peaks in Mn.

| Solid | Temperature °C | Reduction Signal |
|-------|----------------|------------------|
| Mn | <300 | $Mn_5O_8$ to $Mn_2O_3$ ($Mn^{+4}$ -> $Mn^{+3}$) |
| | 350–450 | $Mn_2O_3$ to $Mn_3O_4$ ($Mn^{+3}$ -> $Mn^{+2}$) |
| | 500–600 | $Mn_3O_4$ a MnO |
| | >600 | Mn spinel phase |

*Catalytic Activity*

In order to establish the ideal catalyst ratio for the CWOP of CV, the effect of the catalyst load was assessed with solid Mn–Cu (1:1). This solid was used to determine a cooperative effect between the metals Mn and Cu for the oxidation of the contaminant. At the same time, it established a possible reduction for raw material, increased the available amount of solid for subsequent tests, and ensured a

good performance in the reaction times. According to the literature, the quantities chosen were 1 g, 0.5 g, 0.25 g, and 0.1 g

As a result, only 0.25 g of catalyst (Figure 4) achieved a degradation of 100% CV in 1 h, selectivity in COT of 74%, and elimination in the COD of 70% (Table 4). The results highlight the great potential of these solids compared with catalysts reported previously in the literature [1,2]. Although with any selected amount of catalyst 100% degradation is obtained, increasing the catalytic dose leads to considerable diminution in degradation times, thanks to the increase of active sites of the solid. Then, the highest efficiency was achieved with 1 g of catalyst in a time of 60 min. However, in consideration of both efficiency and costs, a catalyst amount of 0.25 g was selected, with short reaction times of approximately 110 min, with Mn–Cu catalyst (1:1), achieving lower raw material expense, compared with that used for each reaction. Taking into account the ratio of degradation time and the amount of catalyst, with 75% less catalyst compared to 1 g, the increase in the removal time was 45%. Due to these results, a value of 0.25 g was established as the amount of catalyst in the subsequent reactions.

The catalytic wet peroxide oxidation of CV using 0.25 g of catalyst is shown in Figure 5, where the activity was significantly influenced by the mixed oxide synthesis method, and a cooperative effect between the two metals was observed. The results showed that the catalytic activities of the sample Mn–Cu (1:2) were higher than those of the samples of Mn–Cu (1:1) > Mn–Cu (2:1) > Cu > Mn, with degradation times of only 80 min, presenting a clear influence of the molar ratio on the catalytic activity. In this way, a cooperative effect was evidenced, since there was a synergic phenomenon with the molar ratio, and a correlation in which the use of mixed active phases also optimized the physicochemical characteristics and the catalytic properties, in comparison with the system exclusively of Mn or Cu. Additionally, a clear influence was evidenced in shorter times of degradation of the CV in those solids whose Cu ratio was higher, compared to those that had Mn, for which Cu plays an important role in radicals production.

This behavior is related to the TPR-$H_2$ profiles (Figure 3), where the temperatures with minor reduction, corresponding to the solid of the best catalytic activity, validated the efficiency under moderate conditions proposed for the degradation of the CV dye.

Additionally a catalytic test was done without catalyst in order to determine if only $H_2O_2$/air is enough for the oxidation of CV. The percentage of degradation did not exceed 3%, so the presence of catalyst is necessary, confirming the catalytic character of the process and not the reaction medium. Likewise, to see the feasibility of the technique, a test was generated without the addition of $H_2O_2$ or air, and presented a percentage of degradation greater than 50% in 350 min, three times longer than the result with the oxidizing agent. In consequence, the importance of an oxidizing agent such as air and $H_2O_2$ is evident.

A supplementary analysis was done by UV-Vis spectrophotometry. The changes of the UV-visible spectra are shown in Figure 7, where the behavior of the Mn–Cu catalyst (1:2) is highlighted. The signal decrease at 600 nm showed the rupture of the aromatic rings and the formation of aliphatic and easily biodegradable molecules, as evidenced by the appearance of the peaks at 380 and 220 nm, related with the appearance of intermediate compounds during the reaction.

These results confirmed the decomposition of the molecule and the development of by-products that were analyzed by GC-MS (Figure 6). The results in GC-MS for the different characteristic peaks in the spectra allowed the substances to be classified as oxalic acid, ethylbenzene, toluene, carbamic acid, and ethanol, which are probably due to the triphenylmethane breakdown. All these breaks prove that the CV can be degraded into non-toxic small molecules and ensures the elimination of the contaminant in effluents from the proposed method [33].

Based on previous studies about the degradation of CV, it occurs through N-demethylation, chromophore excision and rupture of the trajectories of the ring structure, until the final formation of the desired products. According to the results, it can be stated that the oxidation route was N-demethylation, as a consequence of the reaction caused by the addition of hydroxyl, decomposition of the conjugated structures, elimination of the benzene ring, and the ring opening in smaller molecules [34].

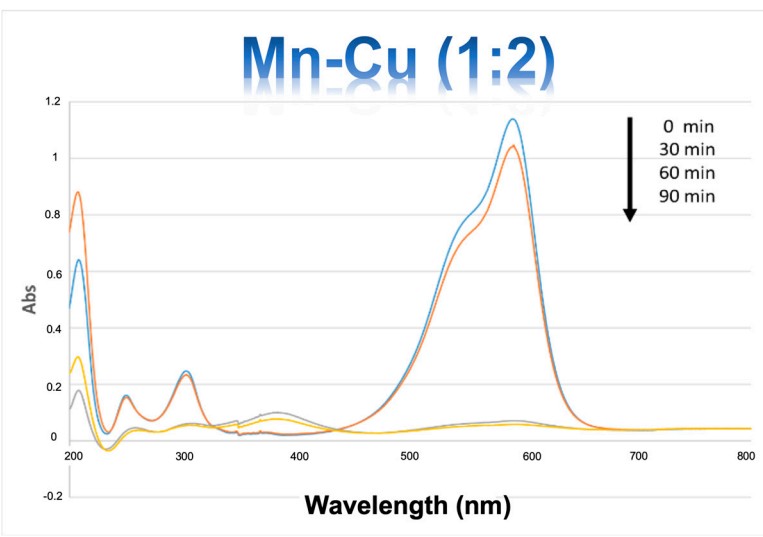

**Figure 7.** UV-Vis spectra for CWPO of CV with Mn–Cu (1:2) catalyst.

## 4. Materials and Methods

### 4.1. Catalyst Synthesis

The mixed oxides were synthesized from hydrated nitrate solutions by the self-combustion method. For the synthesis of Mn–Cu catalysts, $Mn(NO_3)_2 \cdot 6H_2O$, $Mg(NO_3)_2 \cdot 6H_2O$, $Al(NO_3)_3 \cdot 9H_2O$, and $Cu(NO_3)_2 \cdot 3H_2O$ (Merck (Darmstadt, Germany), 95.0% purity), reagents which act as oxidants were used, and glycine $[CH_2NH_2COOH]$ (Merck (Darmstadt, Germany), reagent analytical degree) was used as combustible. Molar ratios for the preparation were Mn = 5, Mn/Cu 1:1 = 1, Mn/Cu 2:1 = 2, Mn/Cu 1:2 = 0.5, and Cu = 5, and a ratio (nitrates/glycine) = 0.8 [35,36]. The aqueous solution obtained was subjected to a slow evaporation process at a temperature of 110 °C, during 120–140 min, and with constant agitation between 200 and 300 rpm, until gel formation. The gel was heated to a temperature of 500 °C to provide a thermal shock and carry out the ignition process, which happened in a matter of seconds. Once the solid was obtained, calcination was carried out at a temperature of 700 °C for 14 h, where the elimination of glycine residues, conformation of the grains, and formation of a well-consolidated crystalline structure were ensured (Figure 8).

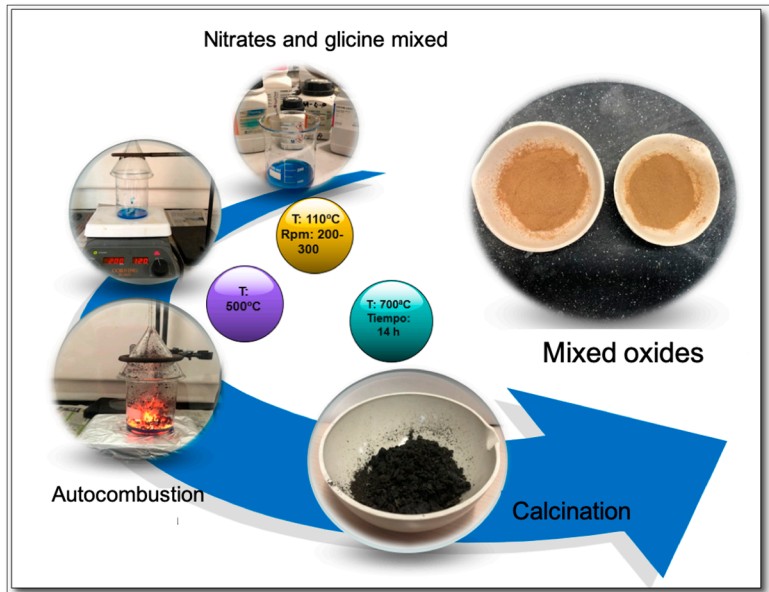

**Figure 8.** Procedure for catalysts synthesis using the autocombustión technique.

### 4.2. Characterization

The catalysts were characterized by X-ray Diffraction XRD (powder sample), using a Panalytical X'Pert PRO MPD (Almelo, The Netherlands) diffractometer equipped with a copper anode ($\lambda = 1.5406$ Å), using an angular velocity of 1°/min and a step size of 0.02° $\theta$ to evaluate the crystalline phases present. The chemical analysis of catalysts was carried out using the X-ray fluorescence XRF technique on a Philips MagiX Pro PW2440 device (PHILIPS / PANALYTICAL, AUSTIN, TX, USA). The oxidative–reductive properties of the materials were measured by means of TPR-$H_2$ in a CHEMBET 3000 QUANTACHROME (Anton-Paar, Boynton Beach, FL, USA) equipment, where a reduction was carried out with a gaseous mixture of $H_2$/Ar, analyzing the processes of reducibility as a function of the temperature. The textural characteristics were determined through $N_2$ adsorption–desorption isotherms using ASAP 2020 Micromeritics equipment (Micromeritics Instrument Corp. Norcross, GA, USA).

### 4.3. Catalytic Evaluation

In order to evaluate the efficiency of the catalyst in the elimination of CV, a semi-batch reactor was used, adding 200 mL of water at a concentration of CV of 50 ppm. It was provided with continuous agitation of 500 rpm, a temperature of 25 °C, constant flow of peroxide of hydrogen 0.1 M 2 mL/h, and air flow of 2 mL/min (Figure 9).

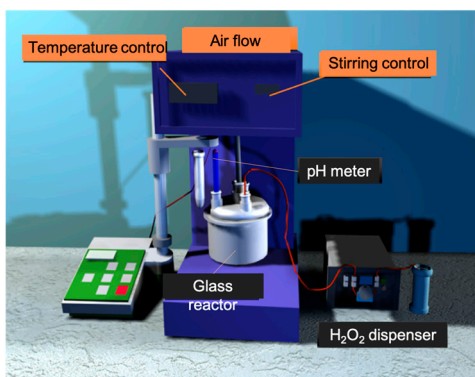

**Figure 9.** Reactor used for the catalytic degradation of CV.

To monitor the elimination of the CV, a Thermo Scientific GENESYS 20 spectrophotometer (Thermo Fisher Scientific Inc, Waltham, MA, USA) was used. The degradation process was measured at a wavelength of 590 nm. In order to determine the selectivity, the amount of total organic carbon (TOC) still present in each reaction time was measured using a Shimadzu Model TOC-L CPH analyzer (Shimadzu, Columbia, MD, USA. The COD was determined through an adaptation of the method approved by EPA 410.4; in a HANNA HI839800 reactor (Hanna Instruments SAS, Madrid, Spain) which determined the amount of oxygen that was required to oxidize the organic matter still present in the samples during the degradation treatment. The reaction was also followed by the GC-MS, which identified the formation of several by-products by the oxidation of the CV during the Fenton-type reaction. Intermediates were identified using a program from the NIST14.L library, with settings around 80%. The reaction mechanisms proposed in this research should be useful for the future application of dye treatment technologies.

### 5. Conclusions

The catalytic activity of the Mn–Cu solids was highly influenced by the association of copper and manganese, exposing a short time of removal of the contaminant (CV) at environmental conditions, where the catalyst Mn–Cu (1:2), with an elimination time of 80 min, was the catalyst with greater efficiency when compared with the other synthesized solids.

The implementation of the method with hydrogen peroxide in heterogeneous phase was efficient in the oxidation of the pollutant, where its catalytic degradation was observed by obtaining by-products (carboxylic acids), until mineralization was achieved.

Additionally, the method showed that no new pollutants were generated. In addition, the mild conditions of temperature and pressure, low airflow and hydrogen peroxide flow are promising for large-scale pilot and industrial implementation, where effluents are handled with high concentrations, being preferred to the techniques exposed in the reference literature, with the aim of reducing pollution and potential toxicity.

The stability of Mn–Cu mixed oxide catalysts during the Fenton reaction process indicated that there was no dissolved metal content in the final water samples, so no additional contamination would be present, excluding some indication of homogeneous catalysis phenomena and indicating, in turn, that the specified operating conditions were suitable within the catalytic system.

**Author Contributions:** Formal analysis, A.P.; Investigation, S.M.; Methodology, D.L.L. and J.A.B.; Project administration, C.A.C.; Resources, S.M.; Supervision, A.P.; Writing—original draft, P.F.R.; Writing—review & editing, A.M.C.

**Funding:** This research was funded by Pontificia Universidad Javeriana. Bogotá. Colombia. Proyecto ID 00007181.

**Conflicts of Interest:** The authors declare no conflict of interest.

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
