# Peer review of "Degradation of Crystal Violet by Catalytic Wet Peroxide Oxidation (CWPO) with Mixed Mn/Cu Oxides"

_catalysts, doi:10.3390/catal9060530_

Round 1
Reviewer 1 Report
Comments on the manuscript entitled “Degradation of crystal violet by catalytic wet peroxide oxidation (CWPO) with mixed Mn/Cu oxides” authored by Ana María Campos et al. (catalysts-506528)
In this paper the authors present the catalytic wet peroxide oxidation of crystal violet dye on Mn/Cu oxides catalysts. The paper presents an interesting activity tests. For this reason this paper could be publishable in Catalysts, however the following comments and suggestions should be addressed in a revised manuscript before it can be reconsidered for publication:
1. Why the authors employ this H2O2 concentration?. Is there any experimental result or data figure to support the initial H2O2 dose? And the temperature? Why?
2. What are the initial values of TOC and COD concentrations (mg/L) of CV dye in the table 4 ?
3. As the authors do not give details of the error in the experiments, it is not possible to know if the differences observed are significant or not; Could give the accuracy of your data and the figures?.
4. Crystal Violet essay in this work is well known to be toxic. In order to evaluate the toxic effects of solutions after treatment, it’s necessary to determine the toxicity if it is possible for authors.
5. It is known that, during the catalytic wet peroxide process of several compounds, polymeric compounds can be formed, Could you explain better this aspect with this compound)?.
6. What is the stability of this catalyst in this reaction?.
7. It is necessary to make a table comparing your results with those of other authors.
8. pH is very important parameter in order to correctly discuss the catalytic performance in the CWPO process. Thus, range of pHs in the treated samples should be clearly disclosed either in the experimental section as well as in the discussion of results. For instance, nature of oxidizing species strongly depends on the pH.
9. Two extra blank experiments are mandatory in order to safely conclude some claims on the overall performance of the catalytic system investigated: (1) only using H2O2 in order to account straightforward attack of the oxidizing agent on the targeted pollutant; (2) a catalytic test in the presence of dissolved Mn or Cu equivalent to the leached concentration measured under optimal conditions of reaction; it is needed to know in what extent homogeneous Fenton would explain degradation of CV.
10. I recommend a full revision of the language in order to avoid further editorial work.
Author Response
Dear Editor
We thank you for your valuable work and through you we wish to thank the evaluators for their suggestions and contributions, which have improved the article.
The document was corrected (paragraph indicated in red) following the suggestions of the evaluators.
Below we respond to each of the comments to the reviewers.
We hope that we have given a satisfactory response to the comments made by the evaluators and that the article can be published.
Best Regards
Alejandro PÉREZ
Reviewer #1:
1. Why the authors employ this H2O2 concentration?. Is there any experimental result or data figure to support the initial H2O2 dose? And the temperature? Why?
In previous work of the group these parameters have been established:
J. G. Carriazo, E. Guelou, J. Barrault, J. M. Tatibouët, S. Moreno; Catalytic wet peroxide oxidation of phenol over Al–Cu or Al–Fe modified clays. Applied Clay Science, Volume 22, Issue 6, June 2003, Pages 303-308
J. G. Carriazo, R. Molina, S. Moreno; A study on Al and Al–Ce–Fe pillaring species and their catalytic potential as they are supported on a bentonite. Applied Catalysis A: General, Volume 334, Issues 1–2, 1 January 2008, Pages 168-172
N. R. Sanabria, M. A. Centeno, R. Molina, S. Moreno; Pillared clays with Al–Fe and Al–Ce–Fe in concentrated medium: Synthesis and catalytic activity. Applied Catalysis A: General, Volume 356, Issue 2, 15 March 2009, Pages 243-249
Alejandro Pérez, Miguel A. Centeno, José A. Odriozola, Rafael Molina, Sonia Moreno; The effect of ultrasound in the synthesis of clays used as catalysts in oxidation reactions. Catalysis Today, Volumes 133–135, April–June 2008, Pages 526-529
2. What are the initial values of TOC and COD concentrations (mg/L) of CV dye in the table 4 ?
The values shown in table 4 of TOC and COD are the percentages of degradation, taking as initial value of 51 and 82 ppm for TOC and COD respectively. Based on these initial values, the effectiveness of these parameters is determined.
3. As the authors do not give details of the error in the experiments, it is not possible to know if the differences observed are significant or not; Could give the accuracy of your data and the figures?.
Indeed, details of errors in the experiments are not shown, due to the great difference in the activity curves and the data shown in the tables. However, as a suggestion the evaluator is denoted in the tables.
4. Crystal Violet essay in this work is well known to be toxic. In order to evaluate the toxic effects of solutions after treatment, it’s necessary to determine the toxicity if it is possible for authors.
As the evaluator points out, the treated molecule presents a degree of toxicity, as well as some of the by-products generated in the oxidation process. The toxicity tests that have been developed are the Microtox Test: inhibition of the luminescence of the Gram-negative bacterium Vibrio fischeri and the Ames Test, observing that the oxidation processes do not generate toxic compounds and the violet crystal is completely eliminated. These trials will be reported in another publication that is under development.
5. It is known that, during the catalytic wet peroxide process of several compounds, polymeric compounds can be formed, Could you explain better this aspect with this compound)?.
It is known that in CWPO reactions there are polymeric compound formation processes, however for our case and for the follow-up of GS-MS were not observed. The oxidation processes were very fast and only small fragments of subcompounds were detailed.
6. What is the stability of this catalyst in this reaction?.
As has already been indicated in the response to reviewer 1, the reuse results form part of another article which is currently undergoing revision, and therefore are not included in this document.
7. It is necessary to make a table comparing your results with those of other authors.
We appreciate the suggestion of the evaluator, however, the few works in literature with this molecule and using mixed oxides in moderate conditions, limit us to indicate them in the form of a bibliography as we have in the article.
Results that clearly indicate the great potential of these solids respect to other catalysts reported in iteratura [20-22], under environmental conditions and low charges of active metal.
8. pH is very important parameter in order to correctly discuss the catalytic performance in the CWPO process. Thus, range of pHs in the treated samples should be clearly disclosed either in the experimental section as well as in the discussion of results. For instance, nature of oxidizing species strongly depends on the pH.
As has already been indicated in the response to reviewer 1, the pH that was handled in this work is approximately 7, and this value is close to the water found in the residues of microbiology laboratories that are being treated with these catalysts and are the results of a work in development. Likewise, the effect of pH with crystal violet has been evaluated and it has been observed that mineralization is facilitated between pH 6-8.
9. Two extra blank experiments are mandatory in order to safely conclude some claims on the overall performance of the catalytic system investigated: (1) only using H2O2 in order to account straightforward attack of the oxidizing agent on the targeted pollutant; (2) a catalytic test in the presence of disolved Mn or Cu equivalent to the leached concentration measured under optimal conditions of reaction; it is needed to know in what extent homogeneous Fenton would explain degradation of CV.
As is mentioned in the article: “These results are associated with good redox properties of Mn and Cu that are enhanced when they are together. Likewise, figure 4 shows that the activity is directly related to the presence of a catalyst and not of the medium of reaction (H2O2 and air).” point 1 has already been made. And because there was no leachate of the active phases (Mn or Cu), there is no need to make a comparison with a homogeneous system.
“Concerning to lixiviation during the reaction process, at the end of the catalytic test, the stability of the active phases was evaluated by taking the solutions after the final reaction time and filtering them to separate the solid from the aqueous phase and assess the possible leaching of the catalyst metals during the reaction. The analysis was done by atomic absorption spectrophotometry technique using equipment Agilent 280.
The nonexistence of leaching of metals (Mn and Cu) measured by atomic adsorption, indicate that the active phase is stable under the reaction conditions and avoiding the generation of additional pollutant.”
10. I recommend a full revision of the language in order to avoid further editorial work.
Thanks to the signaling of the two evaluators and verifying the document again, important changes were made
Reviewer 2 Report
This work deals with the preparation of mixed oxide catalysts of Mn and/or Cu in different molar ratios for the CWPO of crystal violet dye, testing the possibility of cooperation between the two metals. The textural properties of the materials were analyzed by different characterization techniques: XRD, XRF, TPR-H2 and adsorption-desorption of N2. The CWPO of the dye has been tested in a wide pH range, opening the always-narrow pH window of the Fenton process. Moreover, the intermediates products identification by GC-MS is very interesting. I recommend its publication in Catalysts Materials after a few points are addressed:
- Line 80: “… with AN extended range..”
- Molar ratios of the oxide mixed catalysts prepared is 5 for the catalyst with only Cu, 5 for the Mn catalyst, 2.5-2.5 for the catalysts Mn-Cu (1:1) but I don’t understand the catalysts (2:1) and (1:2). If the stoichiometric ratios must add 5 and one of them must be the double of the other, then 1.67-3.33 must be the stoichiometric ratios and no 1.25-3.75…
- Line 134, “.. that are enhanceD..”
- Figure 4: Curves must appear with the experimental points as in Figure 5, not only the lines.
- Figure 4: in the legend, units and numbers appear together, separate them.
- Lines 139, 140, separate numbers and units: “0.25 g”, “150 min”.
- Figure caption of Figure 7: CWPO, not CWOP…
- Lines 305-306: since a possible oxidation route is proposed, it would be interesting to include a scheme with the oxidation route easier to follow than text.
- Reusability of the catalysts has not been tested. Authors think the best catalyst obtained, Mn-Cu (1:2), could be reused several times maintaining its catalytic activity? It would be a good final test for this manuscript.
- Although at the end of the Introduction section is reported that CWPO is going to be tested in an extended pH range, I have not found all along the text the effect of pH on CWPO results.
Author Response
Dear Editor
We thank you for your valuable work and through you we wish to thank the evaluators for their suggestions and contributions, which have improved the article.
The document was corrected (paragraph indicated in red) following the suggestions of the evaluators.
Below we respond to each of the comments to the reviewers.
We hope that we have given a satisfactory response to the comments made by the evaluators and that the article can be published.
Best Regards
Alejandro PÉREZ
Reviewer #2:
1. Molar ratios of the oxide mixed catalysts prepared is 5 for the catalyst with only Cu, 5 for the Mn catalyst, 2.5-2.5 for the catalysts Mn-Cu (1:1) but I don’t understand the catalysts (2:1) and (1:2). If the stoichiometric ratios must add 5 and one of them must be the double of the other, then 1.67-3.33 must be the stoichiometric ratios and no 1.25-3.75…
We thank the evaluator for pointing out the error presented in table 1 of molar relationships, what is indicated by the evaluator is correct, the molar ratios used are for (2: 1) is 3.33: 1.67 and the (1: 2) is 1.67: 3.33 , the synthesis of the solids was carried out maintaining the Mn / Cu (1: 1) = 1; Mn / Cu (2: 1) = 2 and Mn / Cu (1: 2) = 0.5 ratios, unfortunately when we tried to write them in a table for a better understanding they were recorded badly. The pertinent corrections were made and the relationships used in the experimental part of the work were left.
2. Figure 4: Curves must appear with the experimental points as in Figure 5, not
only the lines.
- Figure 4: in the legend, units and numbers appear together, separate them.
The relevant corrections were made
3. Lines 305-306: since a possible oxidation route is proposed, it would be
interesting to include a scheme with the oxidation route easier to follow than
text.
Precisely in the indicated reference (34) a very complete scheme of the oxidation route of the crystal violet by the fenton technique is shown
Fan, H.-J., et al., Degradation pathways of crystal violet by Fenton and Fenton-like systems: Condition optimization and intermediate separation and identification. Journal of Hazardous Materials. 2009. 171(1). 1032-1044.
4. Reusability of the catalysts has not been tested. Authors think the best catalyst obtained, Mn-Cu (1:2), could be reused several times maintaining its catalytic activity? It would be a good final test for this manuscript.
We thank the author for the suggestion, however the reuse and scaling assays (5, 10 and 20L) have been left for another article that is in development. We can comment that the catalyst MnCu (1: 2) in the fourth cycle loses 12% of its activity, showing the stability of this one.
5. Although at the end of the Introduction section is reported that CWPO is going to be tested in an extended pH range, I have not found all along the text the effect of pH on CWPO results.
Usually the use of metals such as Fe, Cu have required the pH adjustment to be active in CWPO reactions. For this work, we want to work in a wider range and especially close to the pH of the water to be treated (residues of microbiology laboratories) for a possible scaling using Mn and Cu metals.
Round 2
Reviewer 1 Report
The authors have fully followed the recommendations of reviewers so the amended version is of much more quality and almost ready for publication.